# Degradation of Dynamic Elastic Modulus of Concrete under Periodic Temperature-Humidity Action

**DOI:** 10.3390/ma13030611

**Published:** 2020-01-30

**Authors:** Depeng Chen, Jiajia Zou, Liang Zhao, Shidai Xu, Tengfei Xiang, Chunlin Liu

**Affiliations:** 1School of Architectural and Civil Engineering, Anhui University of Technology, Ma’anshan 243032, China; dpchen@ahut.edu.cn (D.C.); 17855535776@163.com (J.Z.); agdxsd@163.com (S.X.); xiangtf@ahut.edu.cn (T.X.); 2School of Civil Engineering, Southeast University, Nanjing 210000, China; zhaoliang1231512@163.com; 3Institute of Green Building Materials, Anhui University of Technology, Ma’anshan 243032, China

**Keywords:** periodic temperature-humidity action, dynamic elastic modulus of concrete, fatigue damage, degradation model of dynamic elastic modulus, curve fitting

## Abstract

Cracks caused by environmental temperature and humidity variation are generally considered one of the most important factors causing durability deterioration of concrete structures. The seasonal or daily variation of ambient temperature and humidity can be considered periodic. The dynamic modulus of elasticity is an important parameter used to evaluate the performance of structural concrete under periodic loads. Hence, in this paper, the dynamic elastic modulus test of concrete under simulating periodic temperature-humidity variation is carried out according to monthly meteorological data of representative areas (Nanjing, China). The dynamic elastic modulus attenuation pattern and a dynamic elastic modulus degradation model of concrete under periodic temperature-humidity are investigated. The test results show that the dynamic elastic modulus of concrete decreases and tends to be stable under the action of periodic temperature-humidity. Comparative analysis shows that the two-parameter dynamic elastic modulus degradation model is more suitable for describing the dynamic elastic modulus attenuation pattern of concrete under periodic temperature-humidity action than the single-parameter one.

## 1. Introduction

For most structures, durability is an important performance and a decisive factor which influences service performance during the whole life cycle [1,2]. Environmental impact is generally considered one of the most important factors causing durability deterioration of concrete structures and the performance of concrete under ambient conditions has attracted more and more attention in recent decades [3,4]. It is recognized that the durability of concrete structures is strongly influenced by cracks caused by external loading and other environmentally-induced stresses, that is, the volume stability of concrete. The volume deformation of concrete material is essentially the result of changes in moisture and heat and their coupling action [5]. When the stress induced by load stress and temperature-humidity stress exceeds the tensile strength, the concrete will crack and eventually lead to more serious durability deterioration such as carbonization, chloride ion penetration, internal steel corrosion, and so on [6,7]. Environmentally-induced stresses mainly result from temperature and humidity variation, which are the most common ambient actions and change dramatically during the lifetime of a structure [1].

During the construction period and service life of a concrete structure, the structure is always affected by the temperature and humidity variation of the service environment. There are also indoor and outdoor temperature and humidity differences in industrial and civil buildings which produce temperature and humidity differences between the internal and external surfaces of the concrete components. The changing temperature-humidity-induced temperature gradient and humidity gradient inside of the concrete will cause internal stress and deformation cracking of the concrete structure [8], especially for mass concrete, large-area concrete (such as airport runways, pavement, and rigid waterproof roofs) and super long structure concrete, which is of remarkable significance for the serviceability and durability of concrete structures.

The seasonal or diurnal variation of environmental temperature and humidity can be described by periodic environmental temperature and humidity [9,10]. Compared with non-periodic effects, periodic temperature and humidity produces different stress effects on concrete material and structure, and the mechanism of deformation and cracks will change.

Domestic and foreign scholars have done a lot of research on damage accumulation and performance deterioration of concrete material under temperature-humidity, but most of their works focus on the deterioration of concrete material under freeze-thaw cycles [11,12,13,14,15,16,17], the interaction of freeze-thaw cycles and other factors [18,19,20,21,22], or high-temperature environments [23,24,25,26,27,28]. In fact, besides the long-term uninterrupted freeze-thaw cycles or high temperature effects in special service environments, most concrete is subjected to cyclic moisture-heat effects determined by the local natural environment. As part of a single cycle, freeze-thaw or high temperature processes contribute a lot to deterioration, but the adverse effects of other wet and hot stages on concrete materials cannot be ignored. Thus, researchers have begun to explore the effects of environmental temperature and humidity on concrete workability and setting time [29], internal temperature-humidity response [30,31], volume deformation [32,33,34,35], and thermal durability [36]. However, in the process of constructing the temperature-humidity environment needed for testing, most tests are not based on the actual meteorological conditions in the area but simply formulate the range of temperature-humidity changes artificially, which last for a short time. In fact, concrete material and concrete structures in engineering applications are within complex and changeable meteorological environment. Hence, it is more practical and valuable to construct an experimental environment based on actual meteorological data and carry out experiments in such an environment.

From the perspective of structural design, the compressive strength and elastic modulus of concrete are the most important performance indicators of concrete. These parameters are usually determined, according to the standard procedure, by the uniaxial compression of cylindrical or cubic specimens, and are used as a reference. However, it is not always possible to use these methods in practice, due to their destructiveness and time-consuming nature, and their consumption of numerous specimens [37]. By comparison, the dynamic modulus of elasticity, which is the stress-strain ratio under vibratory conditions and which can be obtained by non-destructive tests, is an important parameter with which to evaluate the performance of structural concrete under periodic loads [38,39]. Non-destructive tests are a kind of low-cost test in which the evaluated component will not be damaged, meaning the test can be repeated. The resonant frequency, ultrasonic pulse velocity (UPV), and surface waves methods are three commonly used non-destructive testing techniques employed to determine the dynamic modulus of elasticity [39]. The UPV test is widely used to evaluate the dynamic modulus of elasticity of concrete structures due to its simplicity, versatility, and repeatability [40].

The dynamic elastic modulus is often used as the damage variable used to characterize the deterioration degree of concrete under the action of sulfate attack, freeze-thaw, and carbonation [2,12,14,18,19,41,42,43]. Some researchers have taken the loss of the relative dynamic modulus of elasticity as the damage variable when investigating the deterioration of concrete under different conditions [2,18,19,41,42,43].

Although there are many works on the damage accumulation and performance degradation of concrete under the action of temperature and humidity, most of them have been carried out for the freeze-thaw cycle. Research on the performance degradation and fatigue effect of materials under the quasi-periodic variation of temperature and humidity in a normal service environment has not attracted enough attention. It is an important factor that cannot be ignored for the performance degradation and durability of mass and overlength concrete structures in areas with large annual or daily temperature variation. Hence, this paper aimed to grasp the dynamic elastic modulus attenuation pattern and the fatigue damage accumulation model of concrete under periodic temperature and humidity. To this end, the dynamic modulus of an elasticity test of concrete with different strength grades under simulated periodic temperature-humidity variation is carried out according to monthly meteorological data of representative areas (Nanjing, China). The dynamic modulus of an elasticity attenuation pattern and the fatigue damage accumulation model of concrete under periodic temperature-humidity are investigated.

## 2. Experiment

### 2.1. Raw Materials and Mix Proportions

Three kinds of concrete with the strength grades C20, C30, and C40 were chosen to carry out a dynamic elastic modulus test under simulated periodic temperature-humidity variation. Chinese standard ordinary Portland cement (PO, similar to Type I cement according to the standard of The American Society for Testing and Materials ASTM C150/C150M-12 [44]) and composite Portland cement (PC, similar to ASTM C150 Type IV cement), according to Chinese standard GB175-2007 [45], produced by a local manufacturer were adopted. Ordinary Portland cement with the strength grade of 42.5MPa (PO42.5) was used to prepare C30 and C40 concrete and early strength Composite Portland cement with the strength grade of 32.5MPa (PC32.5R) was used to prepare C20 concrete. The fine aggregate was natural river sand with a maximum size of 4.75 mm and a fineness modulus (FM) of 2.42. A coarse aggregate of limestone with a diameter of 5–25 mm was used in this study. Common tap water was used as mixing water.

The chemical compositions of the cements PO42.5 and PC32.5R are shown in Table 1. The mix proportions of concrete are shown in Table 2.

### 2.2. Specimen Preparation and Curing Conditions

Cement, river sand, and coarse aggregate were added into 40 L compulsory mixer according to the mix proportion of concrete and stirred for one minute. After this, water was added and the mixture was mixed and stirred for 2 min. Immediately after mixing was finished, the fresh concrete mixture was placed into molds to prepare the specimens. The molds, after being poured full with fresh concrete mixture, were placed on a concrete vibration table and vibrated for one minute to make the concrete compact. All the specimens of the same mix proportion were prepared from the same batch of concrete. According to the Chinese standard GB/T 50082-2009 [46], the specimen size for a dynamic elastic modulus test is 100 × 100 × 400 mm; the specimens were demolded after 24 h and then placed in a standard moist room (temperature 20 ± 2 °C and relative humidity over 95%) for 28 days to be tested.

### 2.3. Periodic Temperature-Humidity

According to the temperature and humidity data of Nanjing City selected from a data set of annual and monthly ground values of China (1981–2010) provided by the national meteorological science data sharing service platform of the Meteorological Data Center of China Meteorological Administration, a simulated environmental periodic temperature and humidity variation pattern was determined for a dynamic elastic modulus test of concrete. The meteorological data of Nanjing City are listed in Table 3.

According to the meteorological data of Nanjing City, the simulated periodic temperature-humidity parameters set for the equipment of the Temperature and Humidity Alternating Test Chamber were as follows: (1) The cycle duration was 24 h, divided into 24 sections, with each section lasting 1 h; (2) the monthly average maximum temperature and the monthly average minimum temperature from January to December, which were used as the test environmental temperature, were assigned to sections 1–24, individually; and (3) the monthly average relative humidity from January to December was assigned to every two sections as the test environmental relative humidity.

The periodic temperature-humidity experimental parameters for the dynamic elastic modulus test of concrete, according to the meteorological data of Nanjing City, are shown in Figure 1.

The temperature-humidity conversion time between each section was not included in the duration of each section. The next section started after the equipment reached the set temperature and humidity at the fastest speed.

A Programmable Temperature and Humidity Alternating Test Chamber (ETE-GDJS-015L), which is shown in Figure 2, was used to provide the simulated periodic temperature-humidity in the dynamic elastic modulus test of concrete.

During the test, due to the time and accuracy deviation between the temperature-humidity inspection instrument and the temperature-humidity probe built in the temperature-humidity alternation test chamber, the actual humidity and temperature conditions in the alternation test chamber were not consistent with the set temperature-humidity parameters. Hence, humidity and temperature parameters measured by the temperature-humidity inspection instrument were selected.

### 2.4. Test Methods

The dynamic elastic modulus of the concrete specimens under cyclic temperature-humidity action was measured and recorded after each temperature-humidity cycle. The measurement of the dynamic elastic modulus was conducted using a resonant frequency method according to the Chinese standard GB/T 50082-2009 [46]. The instrument used in this test was a dynamic modulus tester with an adjusted range of 100 Hz to 20,000 Hz, as shown in Figure 3.

The side perpendicular to the specimen molding surface was selected as the test surface and the middle point was marked as the measurement point, which avoided visible holes or cracks. The frequency range selected was from 1000 to 3000 Hz according to experience and the results of many dynamic elastic modulus tests on concrete.

After curing for 28 days, the specimens were put into the temperature and humidity alternating test chamber and the dynamic elastic modulus of concrete was measured after each cycle. After changing the excitation frequency to make the specimen reach the resonance state, the resonance frequency displayed at this time was measured, and each measurement was repeated more than two times. When the difference between two consecutive measurements did not exceed 0.5% of the arithmetic average of the two measurements, the average values of the two resonance frequencies was taken as the fundamental frequency of the specimen. The dynamic elastic modulus of concrete was calculated according to Equation (1) [46], i.e.,
(1)$$E_{d}=13.244\times 10^{-4}\times WL^{3}f^{2}/a^{4}$$
where *E_d_* is the dynamic elastic modulus of concrete (MPa), *a* is the length of the cross section of the specimen (mm), *L* is the length of the specimen (mm), *W* is the weight of the specimen (kg), and *f* is the fundamental vibration frequency (Hz).

The average value of the measured results of the dynamic elastic modulus of the three specimens in a group was taken as the final measurement value. In general, the calculation should be accurate to 100 MPa.

Three specimens in a group were also weighed with an electronic balance (resolution of 0.01 g) and the weight of each specimen was recorded once after each temperature-humidity cycle.

The weight loss rate (*M_l_*) of each group was able to be calculated from the average weight of three specimens by Equation (2), i.e.,
(2)$$M_{l}=\left(m_{0}-m_{n}\right)/m_{0}\times 100\%$$
where *M_l_* is the weight loss rate of the concrete specimen (%), *m*_0_ is the weight of the concrete specimen before the test, and *m_n_* is the weight of the concrete specimen after *n* times of periodic temperature-humidity cycles.

## 3. Experimental Results and Discussion

### 3.1. Experimental Results

The resonant frequency and weight of different specimens were measured before the test and after each cycle of periodic temperature-humidity in the dynamic elastic modulus test of concrete with the strength grades C20, C30, and C40. The average resonant frequency and average weight of the three specimens in one group were calculated and are listed as original data in Table 4.

There have been few studies on the damage of concrete, which can be evaluated by the weight loss and relative dynamic modulus of elasticity of concrete specimens under periodic temperature and humidity; the research results of concrete damage under freeze-thaw cycles in the study of concrete frost resistance were used in comparative analysis in this paper. According to the periodic cycle of temperature-humidity in this paper and the freeze-thaw test cycle in other works, equivalent fatigue action duration can be used to facilitate comparative analysis of damage parameters in different studies. The basic information of relevant research used in this paper is shown in Table 5.

### 3.2. Weight Loss Rate

The weight loss rate of the concrete specimens was able to be calculated using Equation (2). The weight loss rates in the dynamic elastic modulus test of concrete under periodic temperature-humidity, accompanied by the results of frost resistance research from relevant references are shown in Figure 4.

Figure 4 shows that the weight loss rate of concrete under periodic temperature-humidity increases continuously in a non-linear fashion and that later, the growth rate slows down. Shotcrete under freeze-thaw cycle action shows a similar non-linear increasing trend, but the mass loss rate is relatively small. For ordinary concrete, the weight of the concrete specimen under freeze-thaw cycle action increases in early action duration, and decreases later. For air-entrained concrete, according to Zhao’s research [47], weight loss increases linearly.

Such results should be related to the consistent immersion of concrete specimens in water in the freeze-thaw test. Concrete pores, for example those in air-entrained concrete, play an important role in influencing the weight loss rate of concrete under freeze-thaw cycles. The weight loss rate is also influenced by the strength grade of concrete and decreases as the strength increases.

### 3.3. Relative Dynamic Elastic Modulus

As we all know, the dynamic elastic modulus correlates well with the strength and elastic modulus of concrete. The dynamic elastic modulus of concrete, which can be obtained by non-destructive testing, is suitable to be used as a damage variable to analyze the attenuation pattern of concrete mechanical properties under periodic temperature-humidity action.

After each periodic temperature-humidity cycle, the dynamic elastic modulus (*E_d_*) of C20, C30, and C40 concrete was able to be calculated using Equation (1). To evaluate the attenuation pattern of the dynamic modulus of elasticity of concrete with different strength grades under cyclic temperature-humidity action, the relative dynamic elastic modulus is able to be defined as [43]
(3)$$E_{dr}=E_{n}/E_{0}\times 100\%$$
where *E_dr_* is the relative dynamic elastic modulus of concrete (%), *E*_0_ is the initial *E_d_* of concrete, and *E_n_* is the *E_d_* of concrete after *n* times of periodic temperature-humidity cycles.

The calculated relative dynamic elastic modulus, which was used to evaluate the degradation model of the dynamic elastic modulus of concrete under periodic temperature-humidity action, is listed in Table 6.

The relative dynamic elastic modulus of concrete with three strength grades under periodic temperature-humidity, accompanied by the results in frost resistance research from relevant references, is shown in Figure 5.

Figure 5 shows that periodic temperature-humidity action decreases the dynamic elastic modulus of concrete; however, it also shows that the relative dynamic elastic modulus tends to be stable after more cycles of periodic temperature-humidity variation. The relative dynamic modulus of C40 concrete is obviously higher than that of C30 and C20 after more cycles of periodic temperature-humidity, being over 12% greater.

Under the action of a freeze-thaw cycle, the relative dynamic elastic modulus of ordinary concrete decreases almost linearly, while that of air-entrained concrete decreases non-linearly. With the increase in fatigue duration, according to Cao’s research [48], the relative dynamic elastic modulus of air-entrained concrete is stable at first, and then decreases sharply.

Such results should be related to the strength grade of concrete and should improve the dynamic elastic modulus as the strength increases. Concrete pores, especially for air-entrained concrete, also play an important role in the dynamic elastic modulus of concrete.

## 4. Degradation Model of Dynamic Elastic Modulus Based on Fatigue Damage Theory

### 4.1. Fatigue Damage Evolution Equation

The causes of structural damage are various, and different damage variables and evolution equations can be selected according to diverse damage mechanisms. Even if the same damage process is analyzed, different damage evolution equations will be obtained by using different damage variables.

Fatigue damage is a process of micro crack and micro pore initiation and expansion caused by irreversible evolution of the material structure. This irreversible evolution process directly affects the macro performance of materials. On the basis of continuous damage mechanics, Chaboche [49,50] thought that the damage rate, expressed in terms of cycles N, could be expressed
(4)dD=f(σ, ε, D, M, ⋯)dN
where *D* is the damage variable (maybe the relative dynamic elastic modulus of concrete), σ is the stress, ε is the strain, *M* is the material correlation constant, and *N* is the number of fatigue cycles.

Many other parameters can be considered within Equation (4) according to the problems studied. The damage rate in Equation (4) depends on the present damage state, permitting a non-linear evolution under periodic loading. The function *f* has inseparable variables *N* and *D* in order to describe non-linear damage accumulation and sequence effects [49].

When the loading parameters considered are the maximum stress and mean stress in each cycle, a common general form of the fatigue damage accumulation model is obtained [49,51], i.e.,
(5)$$dD=D^{\alpha \left(\sigma_{M},\sigma_{m}\right)\;}{\left[\frac{\sigma_{M}-\sigma_{m}}{M\left(\sigma_{m}\right)}\right]}^{\beta }dN$$
where β is the material constant, $\sigma_{M}$ is the maximum stress, $\sigma_{m}$ is the mean stress caused by the fatigue load, $M\left(\sigma_{m}\right)$ is the function of mean stress under periodic action, and $\alpha \left(\sigma_{M},\sigma_{m}\;\right)$ is a function of the maximum and mean stress under periodic action.

Lemaitre and Plumtree have proposed the following equation to describe the fatigue damage evolution and defined the form of the function *f*(D) when studying the creep fatigue failures [52], i.e.,
(6)$$\frac{dD}{dN}={\left[\frac{\sigma_{M}-\sigma_{m}}{B\left(\sigma_{m}\right)\;\left(1-D\right)}\right]}^{\beta }f\left(D\right)$$
where f(D) is the function of the damage variable described by Equation (7), accordingly, and $B\left(\sigma_{m}\right)$ is the function of mean stress under periodic action which can be a constant for a certain material.
(7)$$f\left(D\right)={\left(1-D\right)}^{-p}$$
where *p* is the material constant.

Equation (8) for the fatigue damage equation can be obtained by simple substitution of Equation (7) into Equation (6), followed by transformation.
(8)$$dD=\;{\left(1-D\right)}^{-\left(\beta +p\right)}{\left[\frac{\sigma_{M}-\sigma_{m}}{B\left(\sigma_{m}\right)}\right]}^{\beta }dN$$

According to Shang’s research [53], if the energy dissipation characteristics of fatigue damage are combined with the continuous damage theory, Equation (4) can be written as
(9)$$dD={\left(1-D\right)}^{\alpha \left(\Delta \varepsilon /2,\sigma_{m}\right)\;}{\left[\frac{K{\left(\Delta \varepsilon /2\right)}^{n}}{M_{0}\left(1-b^{\prime }\sigma_{m}\right)}\right]}^{\beta }dN$$
where *M*_0_, $b^{\prime }$*,* and *K* are material constants, Δε is the change in strain before and after the periodic action, *n* is the number of periodic actions cycles (in this paper it is the number of periodic temperature-humidity cycles), and $\alpha \left(\Delta \varepsilon /2,\sigma_{m}\right)$ is a function of stain change and mean stress under periodic action.

The above models, Equations (8) and (9), are consistent with the fatigue failure mechanism and are more in line with the conditions of multi-axial and complex load action, such as freeze-thaw fatigue. However, the models are inconvenient to apply because of the inclusion of many parameters which are difficult to obtain. Obviously, the fatigue damage evolution in the form of Equations (8) and (9) can be simplified by considering the material characteristics to be constant for a certain material. Hence, Equations (8) and (9) can be simplified into Equation (10) according to the actual freeze-thaw conditions [54].
(10)$$dD={\left(1-D\right)}^{M_{1}\;}\times M_{2}dN_{\;}$$
where *M*_1_ and *M*_2_ are constants relating to the materials and fatigue load experimental conditions, etc., which can be obtained via fitting analysis of the experimental results.

Both sides of Equation (10) are integrated at the same time and then combined with the boundary conditions of *D* = 0 when *N* = 0. Equation (10) is transferred to the following form [54]:(11)$$D=1-e^{-M_{2}N}\;\left(M_{1}=1\right)$$
(12)$$D=1{-}_{\;}{\left[1-M_{2}N\left(1-M_{1}\right)\right]}^{\frac{1}{1-M_{1}}}\;\left(M_{1}\neq 1\right)$$

Equations (11) and (12) are empirical and cannot deeply describe the macroscopic behavior and reflect the dissipation mechanism of fatigue damage propagation [55]. The empirical equations can simplify the fatigue damage analysis and the constants can be obtained by fitting of the corresponding experimental results.

### 4.2. Degradation Model of the Dynamic Elastic Modulus

If the dynamic elastic modulus loss rate is taken as the damage variable in the analysis of damage and the degradation model of the dynamic elastic modulus of concrete under periodic temperature-humidity action, after substituting *D* = 1 − *E*_n_/*E*_0_ according to Equation (3) into Equations (11) and (12), we can obtain the degradation model of the dynamic elastic modulus of concrete under periodic temperature-humidity action as
(13)$$E_{dr}\left(N\right)=\frac{E_{n}\left(N\right)}{E_{0}}=e^{-M_{2}N}\;\left(M_{1}=1\right)$$
(14)$$E_{dr}\left(N\right)=\frac{E_{n}\left(N\right)}{E_{0}}{=}_{\;}{\left[1-M_{2}N\left(1-M_{1}\right)\right]}^{\frac{1}{1-M_{1}}}\;\;\;\;\;\;\left(M_{1}\neq 1\right)$$
where $E_{dr}\left(N\right)$ is the function of periodic temperature-humidity times, describing the relative dynamic elastic modulus of concrete under periodic temperature-humidity action and $E_{n}\left(N\right)$ is the function of periodic temperature-humidity times, describing the dynamic elastic modulus of concrete after *N* times of periodic temperature-humidity cycle.

### 4.3. Fitting and Comparative Analysis

#### 4.3.1. Fitting Analysis Method

Firstly, the Curve Fitting Tools (cftools) in MATLAB software were used to fit the first eleven relative dynamic elastic modulus data (including the initial and former 10 experimental results of the relative dynamic elastic modulus under periodic temperature-humidity action) according to the relative dynamic elastic modulus degradation model (Equations (13) and (14)), and the fitting parameters (M1 and M2) in Equations (13) and (14) were obtained.

Then, the relative dynamic elastic modulus obtained from the last five experimental results of the dynamic elastic modulus test under periodic temperature-humidity action was compared with the fitting curve of the relative dynamic elastic modulus degradation model (the relevant part of the experimental data), and the rationality of the attenuation model was evaluated by correlation between the curve and experimental results.

#### 4.3.2. Fitting Parameters

The degradation models of the relative dynamic elastic modulus (Equations (13) and (14)) were used as custom equations to fit the initial and former 10 experimental results of the relative dynamic elastic modulus under periodic temperature-humidity action via cftools in MATLAB.

The initial and former 10 relative dynamic elastic moduli obtained from the experimental results and the fitting results of C20, C30, and C40 concrete are shown in Figure 6. The fitting parameters are listed in Table 7.

The fitting goodness values of the dynamic elastic modulus degradation model of C20, C30, and C40 concrete are listed in Table 8.

The fitting effect and practicability of the fitting function can be evaluated by the fitting goodness analysis, mainly based on the SSE and R-square, as shown in Table 8. If SSE is close to 0 and R is close to 1, the fitting function is considered to be more effective and is able to be used for the fitting analysis of some experimental data. According to the fitting goodness parameters, the two-parameter degradation model (when M1 ≠ 1, as shown in Equation (14)), compared with the single-parameter dynamic elastic modulus degradation model (when M1 = 1, as shown in Equation (13)), can better fit the relative dynamic elastic modulus of concrete with the strength grades of C20, C30, and C40 under periodic temperature-humidity action. In other words, the two-parameter dynamic elastic modulus degradation model is more suitable for describing the dynamic elastic modulus attenuation pattern of concrete under periodic temperature-humidity action.

#### 4.3.3. Degradation Model Evaluation and Prediction of Relative Dynamic Elastic Modulus

The fitting parameters (M1 and M2), which were obtained from fitting the initial and the first 10 relative dynamic elastic moduli by the degradation model of the dynamic elastic modulus under periodic temperature-humidity action as custom equations, were substituted into the two-parameter dynamic elastic modulus degradation model (Equation (14)). The degradation model curves were plotted and compared with the experimental relative dynamic elastic modulus results of the last five cycles, as shown in Figure 7.

From Figure 7, it can be observed that the curve of the dynamic elastic modulus degradation model is consistent with the corresponding relative dynamic elastic modulus of the last five periodic temperature-humidity cycles. This good correlation also proves the rationality and feasibility of the degradation model of the dynamic elastic modulus and the proposed fitting analysis method.

It also can be seen from Figure 7 that the relative dynamic elastic modulus of high-strength-grade concrete decreased fast in the early stage and gradually slowed down under the periodic temperature-humidity action. In the middle and late stages, the relative dynamic elastic modulus of high-strength-grade concrete was significantly higher than that of low-strength-grade concrete.

In addition, the verified dynamic elastic modulus degradation model can also be used to predict the relative dynamic elastic modulus after more instances of cycles under the same periodic temperature-humidity action, which can reduce the actual test times and durance to a certain extent. Moreover, the related fitting analysis methods in this paper can also be applied to analyze and solve other similar problems.

## 5. Conclusions

In this work, a dynamic elastic modulus test of concrete under periodic temperature and humidity was conducted with experimental parameters determined according to the monthly meteorological data of Nanjing City. The dynamic elastic modulus deterioration model of concrete under periodic temperature-humidity action was established based on the theory of fatigue damage and the evolution equation of fatigue damage. The conclusions and future research directions are summarized as follows:
The action of periodic temperature-humidity variation decreases the dynamic elastic modulus of concrete, and the dynamic elastic modulus loss rate tends to be stable after more cycles of periodic temperature-humidity. The dynamic modulus loss rate of C40 concrete is obviously lower than that of C30 and C20 after more cycles of periodic temperature-humidity, and is lower by over 12%.2 The weight loss rate of concrete increases continuously, and, later, the growth rate slows down. The weight loss rate is also influenced by the strength grade of concrete and will decrease as the strength increases. The relative dynamic elastic modulus of high-strength-grade concrete was found to decrease slightly faster in the early stage but was significantly higher than that of low-strength-grade concrete later under periodic temperature-humidity action.Taking the loss rate of the dynamic elastic modulus as the damage variable, the dynamic elastic modulus deterioration model of concrete under periodic temperature-humidity action can be established based on the theory of fatigue damage and the evolution equation of fatigue damage.The two-parameter dynamic elastic modulus degradation model (M1 ≠ 1) is more suitable for describing the dynamic elastic modulus attenuation pattern of concrete under periodic temperature-humidity action than the single-parameter model (M1 = 1).The current research in this paper is insufficient, and further research is required. The service load of the actual concrete structure is also an important factor in the deterioration of concrete performance, so the degradation of the dynamic elastic modulus and damage evolution under the combined action of the service load and periodic temperature-humidity are worthy of further study. The micromechanical mechanism of concrete cracking and elastic modulus degradation under periodic temperature and humidity needs to be further studied by means of microscopic analysis, for example via scanning electron microscopy, and finite element analysis. In addition, temperature-humidity conditions closer to the actual service environment should be considered in further research, for example by using the highest (lowest) daily temperature in a certain month as the representative value of the highest (lowest) monthly temperature. The deformation and failure micromechanism of airport or pavement concrete under the coupling effect of wheel load-temperature-humidity in areas with severe temperature and humidity changes is also worthy of attention.

## Figures and Tables

**Figure 1 materials-13-00611-f001:**
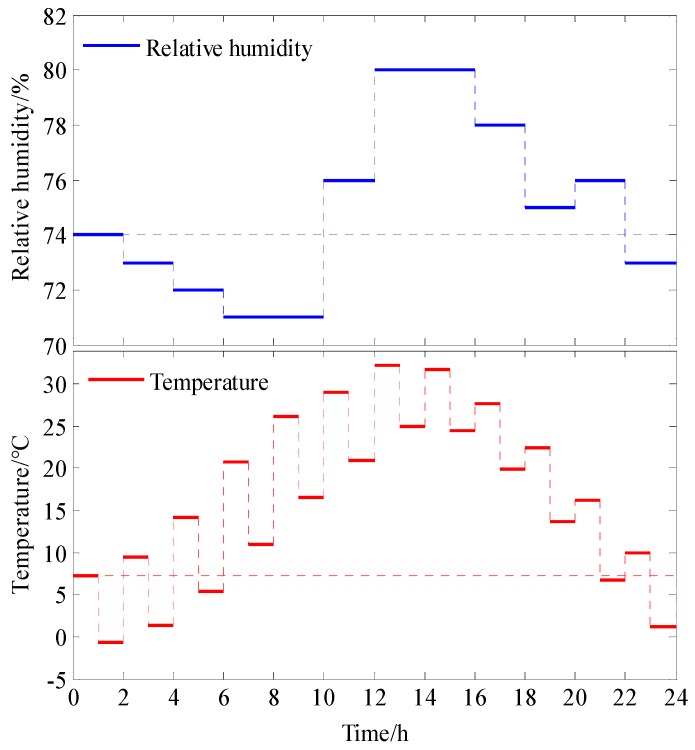
The periodic temperature-humidity experimental parameters.

**Figure 2 materials-13-00611-f002:**
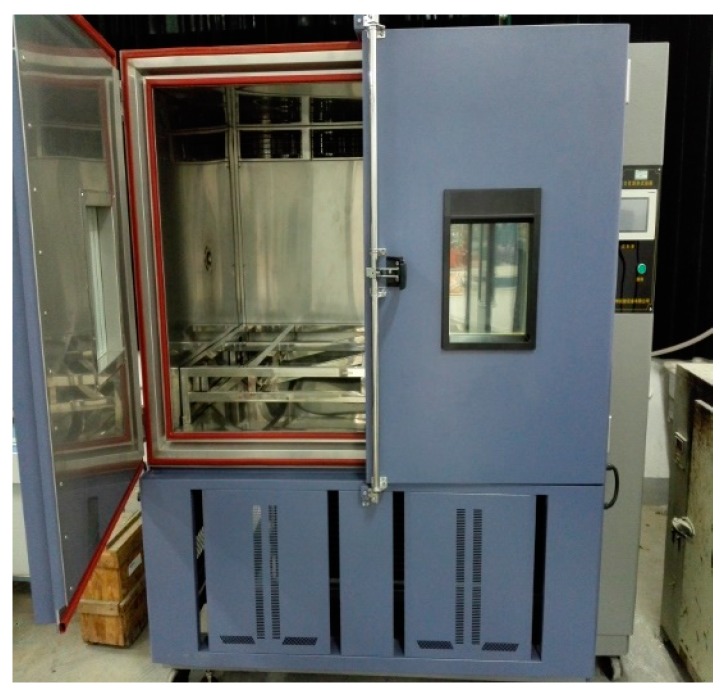
Programmable temperature and humidity alternating test chamber (ETE-GDJS-015L).

**Figure 3 materials-13-00611-f003:**
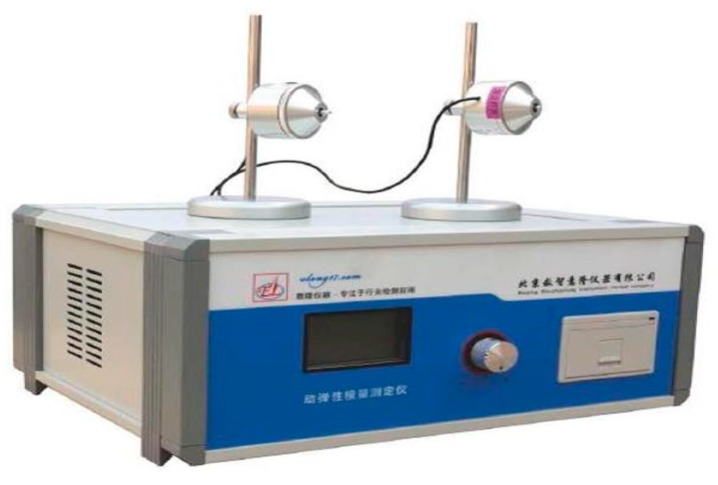
Concrete dynamic modulus of the elasticity tester (DT-W18).

**Figure 4 materials-13-00611-f004:**
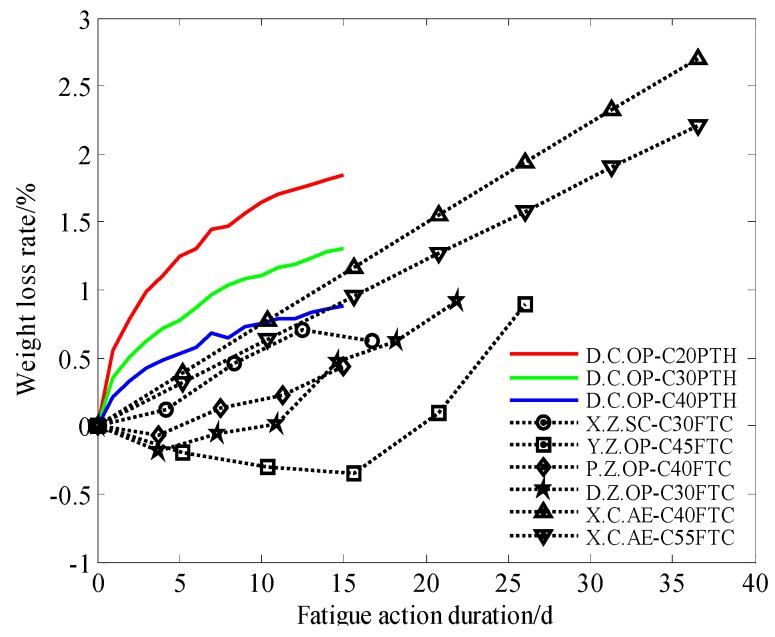
Weight loss rate of concrete under different fatigue actions.

**Figure 5 materials-13-00611-f005:**
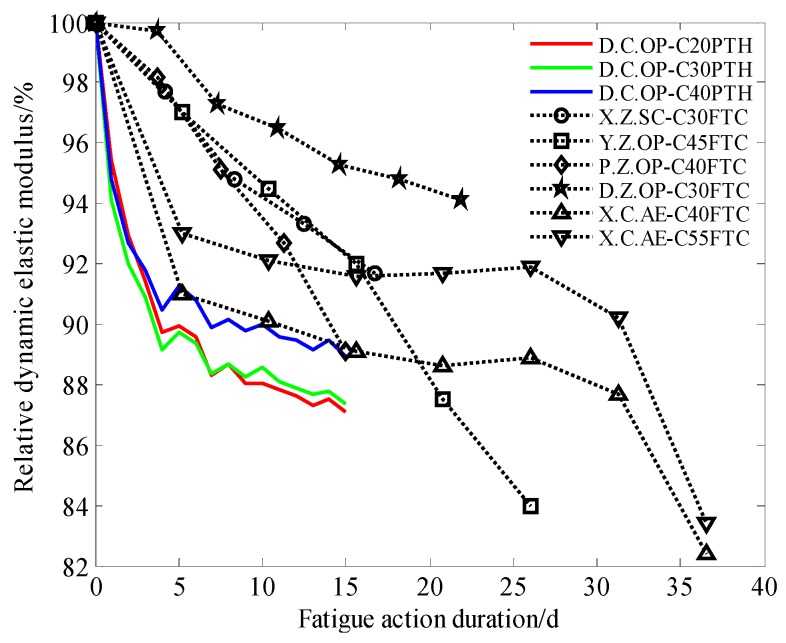
Relative dynamic elastic modulus of concrete under different fatigue actions.

**Figure 6 materials-13-00611-f006:**
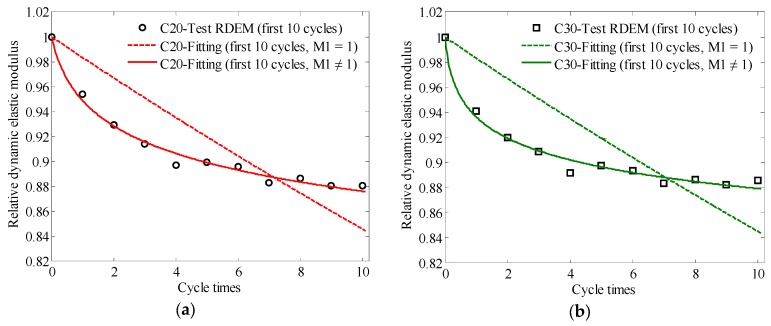
Fitting results of dynamic elastic modulus degradation model based on the initial and first 10 instances of periodic temperature-humidity action: (**a**) C20; (**b**) C30; (**c**) C40; (**d**) two-parameter degradation model for concrete with strength grades of C20, C30, and C40. (Note: C20, C30 and C40 means the concrete with different strength grade of 20 MPa, 30 MPa and 40 MPa; ‘Test RDEM’ means the data is from the relative dynamic elastic modulus test results; ‘Fitting’ means the data is from fitting analysis; ‘M1 = 1’ means the fitting analysis is based on the single parameter model, see Equation (11); ‘M1 ≠ 1’ means the fitting analysis is based on the two-parameter model, see Equation (12), ‘first 10 cycles’ means the data analyzed are form the dynamic elastic modulus test after first 10 times periodic temperature-humidity action cycles).

**Figure 7 materials-13-00611-f007:**
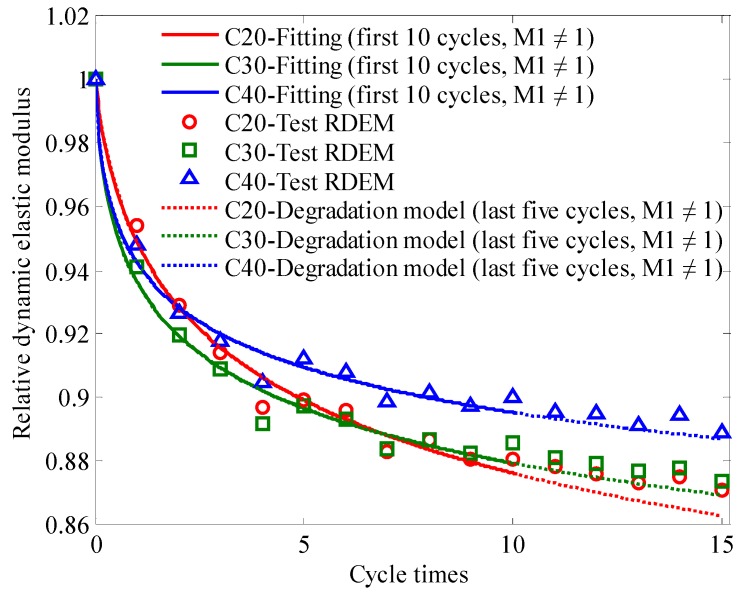
Results from the dynamic elastic modulus degradation model. (Notes: ‘Degradation model’ means the data are calculated based on the dynamic elastic modulus degradation model; ‘last five cycles’ means the data are form the last five cycles of periodic temperature-humidity action in the dynamic elastic modulus test of concrete).

**Table 1 materials-13-00611-t001:** Chemical compositions of cement.

Constituent (wt.%)	SiO_2_	Al_2_O_3_	CaO	MgO	SO_3_	Fe_2_O_3_	Na_2_O	K_2_O	LOI ^1^
PO42.5	32.25	13.04	43.24	1.25	2.1	3.56	0.45	0.85	3.02
PC32.5R	38.23	18.51	25.36	2.76	2.31	3.75	0.22	1.76	6.97

^1^ LOI means loss on ignition of cement

**Table 2 materials-13-00611-t002:** Concrete mix proportions.

Strength	Group	Water-Cement Ratio	Material Dosage (kg)			
			Water	Cement	Sand	Gravel
**C20**	**OC-1**	0.51	195	382	620	1203
C30	OC-2	0.49	195	398	605	1202
C40	OC-3	0.38	195	513	557	1185

**Table 3 materials-13-00611-t003:** Meteorological data of Nanjing City according to a data set of annual and monthly ground values of China (1981–2010).

Months	1	2	3	4	5	6	7	8	9	10	11	12
Average relative humidity (%)	74	73	72	71	71	76	80	80	78	75	76	73
Maximum average temperature (°C)	7.2	9.5	14.2	20.7	26.2	29.1	32.2	31.7	27.7	22.5	16.2	9.9
Minimum average temperature (°C)	−0.7	1.4	5.3	11	16.5	21	24.9	24.4	19.9	13.6	6.8	1.1

**Table 4 materials-13-00611-t004:** Resonant frequency and weight change of concrete specimens.

Cycle Times	Resonant Frequency (Hz)			Weight (g)		
	C20	C30	C40	C20	C30	C40
Initial value	2034.33	2032.67	2100.56	9782.00	9772.00	9736.67
1	1989.11	1973.89	2051.44	9728.00	9736.67	9716.67
2	1966.78	1954.67	2031.22	9704.67	9722.00	9704.67
3	1951.44	1942.67	2020.33	9684.67	9710.67	9696.00
4	1935.11	1924.11	2005.22	9674.00	9701.33	9690.00
5	1937.00	1931.44	2014.11	9660.00	9695.33	9685.33
6	1933.56	1928.56	2010.44	9654.67	9686.67	9680.67
7	1924.00	1918.56	2000.78	9640.00	9678.67	9670.00
8	1926.33	1922.67	2004.89	9638.00	9671.33	9673.33
9	1922.78	1919.44	2000.33	9629.33	9666.67	9666.00
10	1923.56	1921.33	2004.00	9621.33	9664.67	9663.33
11	1920.56	1916.89	1998.89	9616.00	9658.67	9660.00
12	1920.00	1916.78	1997.78	9611.33	9656.00	9659.33
13	1915.56	1912.89	1995.33	9608.67	9650.67	9655.33
14	1918.22	1914.89	1998.33	9604.67	9646.67	9652.67
15	1913.56	1911.78	1993.67	9602.00	9645.33	9651.33

**Table 5 materials-13-00611-t005:** Basic information of relevant research.

Researchers	Concrete Type	Strength Grade (MPa)	Fatigue Cycle (h)	Maximum Number of Cycles	Equivalent Duration (d)	Type of Fatigue Action	Legend in Figure
Zhao, X. [47]	Shotcrete	C30	4	100	17	Freeze-thaw cycle (FTC)	X.Z.SC-C30FT
Zhao, Y. [11]	Ordinary (OP)	C45	5	250	52.1		Y.Z.OP-C45FT
Zhang, P. [17]		C40	3.6	100	15		P.Z.OP-C40FT
Zhang, D. [2]		C30	4	150	21.9		D.Z.OP-C30FT
Cao, X. [48]	Air-entrained	C40	2.5	350	36.5		X.C.AE-C40FT
		C45					X.C.AE-C55FT
Chen, D. (this paper)	Ordinary (OP)	C20	24	15	15	Periodic temperature and humidity (PTH)	D.C.OP-C20PTH
		C30					D.C.OP-C30PTH
		C40					D.C.OP-C40PTH

**Table 6 materials-13-00611-t006:** Relative dynamic elastic modulus of concrete under periodic temperature-humidity.

Number of Cycles	Relative Dynamic Elastic Modulus of Concrete		
	C20	C30	C40
Initial Value	1	1	1
1	0.9541	0.9412	0.9480
2	0.9293	0.9198	0.9267
3	0.9144	0.9090	0.9179
4	0.8972	0.8917	0.9048
5	0.8992	0.8975	0.9124
6	0.8960	0.8934	0.9080
7	0.8829	0.8838	0.8990
8	0.8867	0.8867	0.9018
9	0.8805	0.8826	0.8976
10	0.8805	0.8858	0.9001
11	0.8785	0.8811	0.8957
12	0.8762	0.8791	0.8949
13	0.8732	0.8768	0.8916
14	0.8753	0.8779	0.8946
15	0.8709	0.8738	0.8892

**Table 7 materials-13-00611-t007:** Fitting parameters of the dynamic elastic modulus degradation model.

Concrete	M1 ≠ 1		M2 (M1 = 1)
	M1	M2	
C20	26.54	0.1103	0.01672
C30	36.11	0.2566	0.0168
C40	44.53	0.2798	0.0145

**Table 8 materials-13-00611-t008:** Fitting goodness of the dynamic elastic modulus degradation model. Legend: SSE, sum of squares due to error; RMSE: root mean square error; R-Square: coefficient of determination.

Model Types	Concrete Strength	Parameter			
		SSE	R-Square	Adjusted R-Square	RMSE
M1 = 1	C20	0.007366	0.4708	0.4708	0.02714
	C30	0.01057	0.1528	0.1528	0.03251
	C40	0.00865	0.0734	0.0734	0.02941
M1 ≠ 1	C20	0.0001785	0.9872	0.9858	0.004453
	C30	0.0002001	0.984	0.9822	0.004715
	C40	0.0001799	0.9809	0.9788	0.004446

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
