# Peer review of "Degradation of Dynamic Elastic Modulus of Concrete under Periodic Temperature-Humidity Action"

_materials, 2020, doi:10.3390/ma13030611_

Round 1
Reviewer 1 Report
In this manuscript, the authors performed experiments and analyzes different from the existing literatures on dynamic modulus of elasticity attenuation pattern and the dynamic elastic modulus degradation under periodic temperature-humidity action, and proposes a suitable model for dynamic elastic modulus degradation.
I think it is worth publishing to this journal.
Author Response
We thank the reviewers for their careful read and thoughtful comments on previous manuscript. Thanks very much!
Reviewer 2 Report
The manuscript presents an experimental work on the effect of temperature-humidity variation on the degratation of the dynamic elastic modulus of concrete. In particular, the effect of periodic temperature-humidity variations is investigated by means of empirical equations, including an evolution law based on continuous damage mechanics theory of Chaboche.
The topic is interesting and relevant to the Journal. The experimental campaign is well conducted and clearly described. A weak aspect of the manuscript is related to the theoretical background on the actual damage and microcracking mechanisms leading to elastic modulus degratation. In my opinion the manuscript can be accepted for publication provided that the following remarks are carefully considered by the Authors in their revised version of the manuscript:
1. In relation to Table 3, it is not well described if the temperature and humidity data reported in the table accounts for both daily and seasonal excurtions considered for Nanjing city. Please clarify. Also, it would be good to have the data presented in Table 3 in a graphical form;
2. The frequency range adopted of 1-3 kHz should be motivated;
3. L is the length of specimen and not ‘the length of cross section of specimen’ (see the line below eq. 1);
4. Conversely to eq. 1, eq. 2 is not a dimensional equation and hence the physical dimension (g) of the mass m_0 and m_n is not need (note that weight and mass notation is mixed up). The same comment holds true for the text after eq. 3 and eq. 5;
5. The evolution curves of Figs 3 and 4 present the same asympthotic trend, but it seems that for dynamic elastic modulus the asymptote is reached more quicly with respect to the mass case. Please comment on this point;
6. The statement in line 217 (E_dr + E_dl = 1) is trivial. Why not using a single degradation parameter to describe the dynamic elastic modulus as both E_dr and E_dl are dependent one each other?
7. The symbol N is used to describe the ‘change of fatigue times’. The jargon is not appropriate. Do the Authors mean N = number of fatigue cycles?
8. The symbol n is used to describe the ‘times of periodic actions’, see lines below eq. 6. Please clarify;
9. The background related to eq. 6 should be better clarified. Please expand the relevant text;
10. Instead of describe fatigue damage evolution in terms of number of cycles N, one could relate damage to time unit so as to express relevant equations in Sect. 4 as damage rate laws;
11. The non-linear fitting curves of Figs 5 and 6 (where M1 is different from 1) seem to well describe the experimental data in the range 1-15 of temperature-humidity cycles. What happens when one tries to predict the degradation of the dynamic elastic modulus using these fitting curves for higher numbers of cycles?

Author Response
Thank you for your careful read and thoughtful comments on previous manuscript. We have carefully revised the manuscript according to the comments.
Please see details in the attached file.
Chinese Spring Festival is coming, happy Chinese New Year to You!

Reviewer 3 Report
The present paper aimed to grasp the dynamic elastic modulus attenuation pattern and the fatigue damage accumulation model of concrete under periodic temperature and humidity. The study is interesting and falls in the scope of the journal although the results can be intuited without reading the paper. So, the authors must make clear the advance of knowledge that this work entails. The paper is well written and well structured and it is easy to read and understand for readers.
The introduction is well organized and many references about other works are cited.
The methodology carried out is clearly exposed although more probes, at least 3, must have been manufactured for each cycle. With only a probe no conclussions can be obtained.
The results is clearly exposed but the relationship with the existing bibliography is missing. Please, add references and relations with the current state of knowledge.
Author Response
Thank you for your careful read and thoughtful comments on previous manuscript. We have carefully revised the manuscript according to the comments.
Please see details in the attached file.
Thanks a lot!

Reviewer 4 Report
The authors present a research work related to the “Degradation of Dynamic Elastic Modulus of Concrete under Periodic Temperature-Humidity Action” where the dynamic modulus of elasticity attenuation pattern and the dynamic elastic modulus degradation model of concrete under periodic temperature humidity were investigated.
Remarks to the authors:
Please relate how your results can be interpreted in the context of other studies. Where is the case, please argue the obtained results. Please clearly define the novelty of the manuscript. Rewrite properlyLine 85 .And
Line 134 to section 1 to section 24
Line 234 .And
Line 254 .But
Line 284 wren, .And
Line 309 .Then
Future research directions may also be mentioned.Author Response
Thank you for your careful read and thoughtful comments on previous manuscript. We have carefully revised the manuscript according to the comments.
Please see details in the attached file.
Thanks a lot!

Round 2
Reviewer 2 Report
The Authors revised the manuscript according to my review comments.
Reviewer 3 Report
The authors have really improved the paper. So, I can Accept the paper in the present form.